# Experimental nerve block study on painful withdrawal reflex responses in humans

**Oumie Thorell** [1,2]*, **David Mahns**[1], **Jan Otto**[3], **Jaquette Liljencrantz**[4], **Mats Svantesson**[2], **Håkan Olausson**[2], **Saad Nagi** [1,2]

1 School of Medicine, Western Sydney University, Penrith, Australia, 2 Department of Biomedical and Clinical Sciences, Linköping University, Linköping, Sweden, 3 Department of Neurology, University Hospital Schleswig-Holstein Kiel, Kiel, Germany, 4 Department of Anesthesiology and Intensive Care, Institute of Clinical Sciences, Sahlgrenska Academy at University of Gothenburg, Gothenburg, Sweden

* oumie.thorell@liu.se

## Abstract

The nociceptive withdrawal reflex (NWR) is a protective limb withdrawal response triggered by painful stimuli, used to assess spinal nociceptive excitability. Conventionally, the NWR is understood as having two reflex responses: a short-latency Aβ-mediated response, considered tactile, and a longer-latency Aδ-mediated response, considered nociceptive. However, nociceptors with conduction velocities similar to Aβ tactile afferents have been identified in human skin. In this study, we investigated the effect of a preferential conduction block of Aβ fibers on pain perception and NWR signaling evoked by intradermal electrical stimulation in healthy participants. We recorded a total of 198 NWR responses in the intact condition, and no dual reflex responses occurred within our latency bandwidth (50–150 ms). The current required to elicit the NWR was higher than the perceptual pain threshold, indicating that NWR did not occur before pain was felt. In the block condition, when the Aβ-mediated tuning fork sensation was lost while Aδ-mediated nonpainful cooling was still detectable (albeit reduced), we observed that the reflex was abolished. Further, short-latency electrical pain intensity at pre-block thresholds was greatly reduced, with any residual pain sensation having a longer latency. Although electrical pain was unaffected at suprathreshold current, the reflex could not be evoked despite a two-fold increase in the pre-block current and a five-fold increase in the pre-block pulse duration. These observations lend support to the possible involvement of Aβ-fiber inputs in pain and reflex signaling.

## Introduction

Pain involves cognitive, genetic, and psychosocial factors [1–4]. Currently, pain assessment mainly relies on individuals' self-report which has limitations. In the early 20th century, reflexes were extensively studied by Sherrington who observed coordinated muscle movements, such as flexion followed by stepping movements, that correlated with the intensity of noxious stimulation [5]. This phenomenon is often termed the nociceptive withdrawal reflex (NWR) and involves a complex interplay between top-down and bottom-up influences [6–9].

**Data Availability Statement:** All original data are available as an exel-file for replication studies.

**Funding:** This work was funded by the Swedish Research Council (S.S.N.), Knut and Alice Wallenberg Foundation (H.O.), ALF Grants Region Östergötland (S.S.N.), Swedish Medical Society (S.S.N.), and Western Sydney University (D.A.M.). The funders had no role in the study design, data collection and analysis, decision to publish, or preparation of the manuscript. There was no additional external funding received for this study.

**Competing interests:** The authors have declared that no competing interests exist.

It was initially suggested that the NWR may serve as a readout of pain, but several studies have since highlighted that the relationship between pain and reflex is not clearcut [10–16]. Nonetheless, the reflex is a useful tool for monitoring the excitability of nociceptive spinal systems.

The NWR is often reported as consisting of two distinct electromyographic (EMG) responses with an intervening silent period [10, 17, 18]. These responses, RII and RIII, are attributed to different peripheral afferents: RII to large, thickly myelinated afferents with Aβ-fiber conduction velocities (CV) and RIII to smaller, thinly myelinated afferents with Aδ-fiber CVs [19]. It is generally argued that the RII is non-painful, and it is the RIII that represents spinal nociceptive signaling [10, 20, 21]. Thus, it is common practice to exclude the first NWR component from the reflex analysis. However, there is no consensus on where in time the separation between RII and RIII should occur. Using a 90-ms cutoff, for instance, it was found that NWR responses shorter than 90 ms were just as painful as those that were ≥90 ms [22]. Further, the NWR may comprise of single EMG responses occurring at different latencies and stimulation intensities, questioning the involvement of distinct peripheral afferent classes [20, 22].

It was recently reported that human skin is equipped with a specific class of high-threshold mechanoreceptors with Aβ conduction velocities. These receptors encode noxious mechanical stimuli and produce painful percepts when selectively activated through low-current intraneural stimulation [23–26]. This discovery raises the question of whether Aβ inputs contribute to painful NWR signaling in humans. Part of the ambiguity in the literature around latencies is due to the paucity of direct recordings from Aδ afferent fibers in humans, thereby relying on indirect measurements to infer their conduction velocities. Indeed, in animals, the conduction velocity of D-hair afferents is used as a cutoff between Aβ and Aδ populations [23]; however, D-hair afferents have not yet been characterized in humans.

In the current study, we employed preferential Aβ-fiber conduction blocks and tested pain and reflex responses evoked by intradermal electrical stimulation before, during, and upon recovery from the block. Nerve conduction blocks are widely used in the somatosensory field to study the functions of primary afferent fibers [27–29]. In the block condition where tuning fork sensation was abolished while nonpainful cooling remained relatively preserved–readouts of Aβ- and Aδ-fiber activity, respectively–we found that reflex responses at pre-block thresholds could not be evoked. Further, short-latency pain intensity at pre-block electrical thresholds was significantly reduced during the block, with any residual pain sensation having a longer latency. Although pain could be evoked at higher stimulus intensities, the reflex could not be evoked during the block despite considerable increases to the pre-block stimulus intensities and duration, hinting at the potential involvement of Aβ fibers in driving our nocifensive behaviors.

## Methods

### Participants

Twenty-five healthy participants (17 males and 8 females), aged between 18 and 39 years (median age: 23.5 years), took part in this study. The exclusion criteria included neurological and musculoskeletal disorders, skin diseases, diabetes, and the use of pain-relieving or psychoactive medications. The study was approved by the Swedish Ethical Review Authority (dnr 2020–04207). Written informed consent was obtained from all participants before the start of the experiment. The study was conducted in accordance with the Helsinki Declaration. Recruitment period 2021-02-08 to 2021-06-24. Participants were seated comfortably in a chair with the knee flexed to ~130˚.

## EMG recordings

Three self-adhesive recording electrodes (Kendall ECG electrodes, 57x34 mm, Medtronics, USA) were attached to the right tibialis anterior (TA) muscle serving as active, reference, and ground points. EMG recording settings comprised a 1 mV range, 1 kHz low pass filter, 0.3 Hz high pass filter, and 20 kHz sampling rate (LabChart v8.1.16, ADInstruments, Dunedin, New Zealand).

## Intradermal electrical stimulation

Two uninsulated tungsten microelectrodes (FHC Inc., Bowdoin, USA) were inserted just below the metatarsophalangeal joint of the right foot sole, spaced 5-mm apart, to deliver focal electrical stimulation. The microelectrodes had a tip diameter of 5–10 μm and a shaft of 200 μm. Each stimulus trial consisted of 5 square pulses delivered at a frequency of 200 Hz, with a pulse duration of 0.2 ms. In microneurography, short pulse durations have been shown to evoke a pain percept when Aβ nociceptors are selectively activated [24]. Our approach, using needle electrodes, aimed to minimize current spread. Therefore, we chose the shortest pulse duration at which an NWR could still be elicited. These stimuli were generated using a constant current bipolar stimulator (DS8R, Digitimer, Hydeway, UK). In cases where a reflex could not be evoked with a pulse duration of 0.2 ms, a longer pulse duration of 1 ms was used. The interstimulus intervals were varied from trial to trial (at least >6 s) to prevent habituation and/or cognitive suppression of the reflex response. To avoid visual and auditory cues, a partition was placed between the participant and the experimenter, and a silent mouse was used to trigger the stimuli.

## Perception and reflex threshold measurements

Participants were instructed to remain relaxed during the recordings, which were performed without any muscle contraction. Current was slowly increased and decreased in increments of 1–3 mA, until the first (nonpainful) sensation was reported. This was taken as the detection threshold ($DET^{th}$). The same procedure was repeated to establish the minimum current required to evoke a painful percept (pain threshold ($PAIN^{th}$). NWR thresholds ($NWR^{th}$) were determined based on at least two successful trials at the same current. $PAIN^{th}$ and $NWR^{th}$ were established during baseline. Participants were also asked to rate the intensity of pain on a visual analog scale (VAS) ranging from 0 to 10, with 0 representing "no pain" and 10 representing the "worst imaginable pain" (Response meter, ADInstruments, Dunedin, New Zealand). Participants were instructed to move the analog scale only if the sensation was perceived as painful, and they were free to interpret pain according to their individual experiences. Pain qualities were captured using a short-form McGill Pain Questionnaire [30] immediately following $PAIN^{th}$ and $NWR^{th}$ measurements. The pain questionnaire was administered a total of six times, once for each condition (baseline, block, recovery) at pre-block $PAIN^{th}$ and once for each condition at pre-block $NWR^{th}$.

## Reflex analysis

Reflex latencies, Z scores, and pain ratings were analyzed in MATLAB (r2021b, MathWorks Inc, Natick, Massachusetts). Following individual inspection of EMG recordings, a time analysis window of 50-150ms was adopted. Z scores ($> 1$) were calculated as the difference between peak amplitude (50–150 ms post-stimulus onset) and mean baseline amplitude (-0.15 to 0 ms relative to the stimulus onset), divided by the standard deviation of baseline EMG activity. The minimum current required to evoke a reflex response was taken as the $NWR^{th}$. Responses with

latencies exceeding 150 ms after stimulus onset were excluded from the analysis to avoid voluntary/startle responses that can follow reflex elicitation [31].

## Nerve block

An ischemic nerve block progressively affects large myelinated fibers that signal vibration, followed by small myelinated fibers that signal innocuous cold, and finally, unmyelinated fibers that signal warmth sensations [27–29, 32–34]. To induce the block, an air-filled pressure cuff (Riester Gmbh, Jungigen, Germany) was placed over the right ankle and inflated to 300 mm Hg for up to an hour [35]. The block was applied distal to the TA muscle EMG was recorded from.

In order to track the progression of the nerve block, vibration perception (test for Aβ-fiber function) was tested on the foot sole in three ways: *1*. With a tuning fork (128 Hz, American Diagnostic Corporation, NY, USA) using a two-alternative forced choice detection task (2AFC) where participants reported whether the tuning fork was perceived as "vibration" or "no vibration"; *2*. With a punctate Piezo electric stimulator (probe diameter: 1.3 cm, Dancer Design, UK) where participants rated the intensity of vibration (20 or 200 Hz) on an analog scale ranging from 0 ("no vibrating sensation") to 10 ("highest vibrating intensity"); *3*. Using a 3AFC detection task where participants reported whether vibration at 200, 20, and 0 Hz was perceived as "high", "low" or "no" vibration. Participants wore earplugs during vibration tests to prevent auditory cues. When participants could no longer distinguish whether the tuning fork was stationary or vibrating, the blockade of Aβ fibers was considered successful. In addition, it was expected that vibratory stimuli would be rated as less intense during the block, and participants would be unable to discriminate between 20 and 200 Hz frequencies.

To assess Aδ- and C-fiber functions, we conducted simple detection tasks by placing a cold and hot metal rod, which had been immersed in ice and a water bath at 45˚C, respectively, against the metatarsophalangeal joint of the foot and contralateral (intact) foot sole every ~5 minutes. Participants were asked to verbally report what they felt and whether the intensity between the two sites was the same or different. This allowed for frequent testing of thermal sensibility. As soon as the tuning fork sensation was lost, and other vibratory tests were performed, detection thresholds for cooling and warming were measured on the foot sole using the method of limits. The thermode probe had dimensions of 30 x 30 mm (TSA-II, Medoc Ltd., Ramat Yishai, Israel) and the rate of temperature change was 1˚C/s, starting from a neutral temperature of 32˚C [36]. Each modality was tested four times. In the optimal condition where the vibration sense was blocked while temperature senses remained, perceptual responses were tested at least 3 times at $PAIN^{th}$ and $NWR^{th}$ intenisties.

In the pre-block condition, reaction time measurements were conducted 10 times at the $PAIN^{th}$ stimulus current. During this assessment, participants were asked to press a button as soon as they felt a painful sensation. The inter-stimulus intervals were pseudorandomized (mean: 3.7 s, min: 1.1 s, max: 7.5 s). During the block, if any pain was reported by the participants at the $PAIN^{th}$, the reaction time measurements were tested again.

Upon release of the nerve block, vibratory and thermal sensibilities were monitored, and upon recovery to pre-block levels (typically within 20–30 min), $NWR^{th}$ and $PAIN^{th}$ were measured again.

## Control experiment

To confirm that any effect of the nerve block on pain and reflex responses was due to the blockade of peripheral A-fiber inputs rather than central or other factors, in five participants we ran control experiments with the ischemic cuff applied to the contralateral (left) leg. Pain and reflex responses were measured from the standard (right) leg.

## Statistical analysis

The experiment followed a quantitative, repeated-measurement design in which participants were always tested in three conditions: before (Baseline), during (Block), and after (Recovery) of the nerve block. Descriptive statistics and analyses were performed in Prism (9.0.2, GraphPad Software, San Diego, USA). Non-parametrical statistical tests were chosen because of the dataset's medium to small sample sizes, non-normal distribution (as indicated by normality tests), skewed distribution (QQ-plots, skewness, and kurtosis), and/or high standard deviation (SD) in relation to mean values ($>50\%$). Wilcoxon test was used to compare two related groups. Friedman´s test was used to compare multiple related groups and Kruskal-Wallis for multiple non-related groups. Dunn's test was used as a post hoc for multiple comparisons. Spearman's rank was used to investigate correlations with a 95% confidence interval (CI). A p-value of $< 0.05$ was considered statistically significant. The a-priori sample size estimation was based on a pilot study where we observed an effect size (f) of 0.255. We then used a $1-\beta$ error probability (power) of 0.80, and $\alpha$ error probability of 0.05, which gave a total sample size of 27. Post hoc power analysis, based on f(0.27), $\alpha$(0.05), and a sample size of 25, gave a power of 82.6. Sample size and power calculations were performed in G$^*$ Power (open software, v3.1.9.7). Data are shown as median with interquartile range.

## Results

Under intact (baseline) conditions, 198 NWR responses were obtained from 22 out of 25 participants, ranging from 4 to 16 responses per participant. A reflex threshold (NWR$^{th}$) was established for all 22 participants. Three participants were excluded because full experiment could not be performed, resulting in insufficient data. The NWR$^{th}$ corresponded to the lowest current (mA) at which a reflex could be evoked at least twice during the intact condition. The current intensity required to reach $DET^{th}$ was the lowest, followed by $PAIN^{th}$, and finally $NWR^{th}$ (Fig 1A and 1B). Consequently, all reflex thresholds occurred in response to a painful stimulus, with pain intensity ratings ranging from 0.3 to 6 on a VAS of 0–10. Detection, pain, or reflex thresholds did not show any correlation with the age or sex of participants.

A breakdown of individual reflex responses reveals that only 9 out of 198, or less than 5%, were rated as nonpainful, and in no participant were two consecutive reflex responses rated as nonpainful. During the block condition, the NWR$^{th}$ was tested at least 3 times using each participant's individual pre-block current. Further, suprathreshold intensities (up to twice the pre-block current) were tested during the block condition. This was to make sure that any effect on the NWR during the block was not merely due to a shift in NWR$^{th}$, but rather indicated a completion abolition of the NWR. In the recovery condition, the NWR was considered recovered if it was elicited at least once at the pre-block current threshold.

The NWR responses had Z scores ranging from 1 to 61 and occurred at latencies between 65 and 137 ms after stimulus onset (mean latency: 91 ms). Out of these, 80 (40.4%) NWR responses had latencies under 90 ms, a cutoff defining onset of RIII reflexes, as used in prior studies [e.g. 15, 16, 37]. In another study, involving transcutaneous electrical stimulation, dual reflex responses (RII and RIII) were observed in 12% of reflex recordings [22]. In the current study using intradermal electrical stimulation, no instances of dual RII-RIII responses were observed (Fig 1C). In a subset of participants (n = 7), the reflex could only be evoked using a 1-ms pulse duration. However, there were no statistical differences in NWR$^{th}$, reflex latencies, or pain ratings based on pulse duration, hence the data were combined (S1A–S1D Fig).

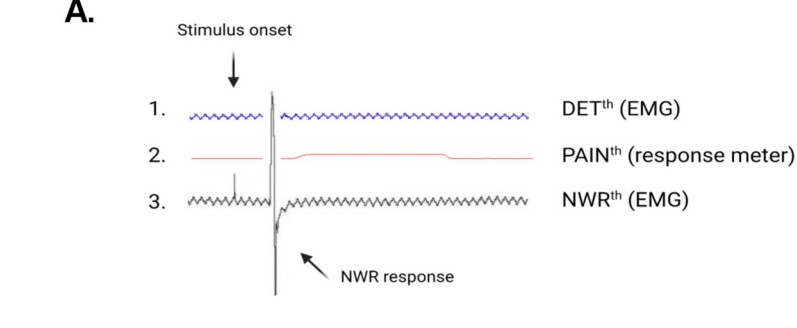

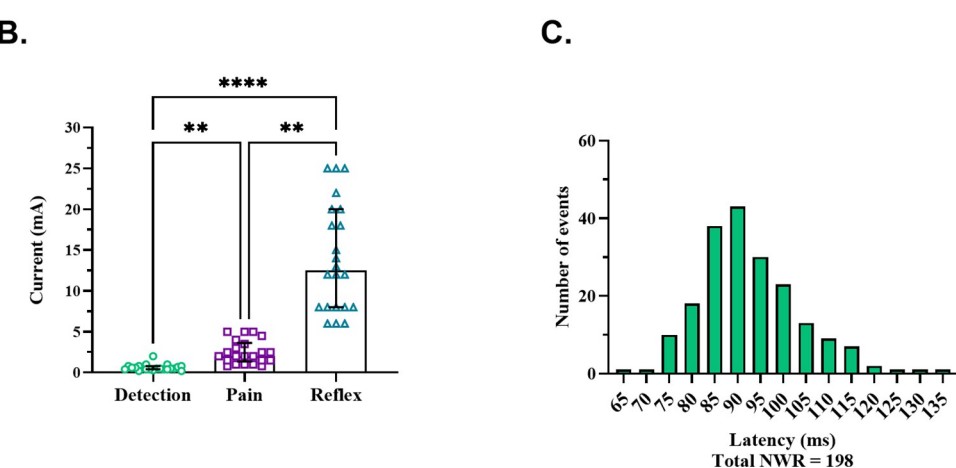

**Fig 1. Characterization of pain and reflex responses evoked by intradermal electrical stimulation. A**. The first trace shows the absence of an EMG response at the nonpainful detection threshold ($DET^{th}$). The second trace shows a pain rating at the pain threshold ($PAIN^{th}$), although no EMG response was detected. The third trace shows an EMG response at the NWR threshold ($NWR^{th}$). **B**. $DET^{th}$, $PAIN^{th}$, and $NWR^{th}$ were significantly different ($DET^{th}$: 0.5 (0.2) mA, $PAIN^{th}$: 2.0 (1.5) mA, $NWR^{th}$: 13.5 (12.0) mA, f(2) = 44.00, p < 0.0001, post hoc test: **P = 0.0027; ****P < 0.0001, n = 22, Friedman test). **C**. A total of 198 NWR responses were recorded with no instances of dual EMG bursts within our latency bandwidth (50–150 ms).

## Preferential block of Aβ fibers

Somatosensory tests were performed to gauge the progression of the ischemic nerve block. During baseline and recovery conditions, participants performed with 100% accuracy in the vibration discrimination tasks (2AFC, 3AFC). During the block for >20 min but <1 hour, participants could no longer distinguish whether the tuning fork was stationary or vibrating. Further, the vibration intensity ratings declined significantly (Fig 2A and 2B), and vibration discrimination was significantly impaired (Fig 2C).

Cold detection thresholds (CDTs) were significantly altered (median difference from baseline = 7.2°C) during the nerve block (Fig 2D). However, in no participant did the mean CDTs shift to the cold pain threshold range (reported as ≤10–14°C) [38], indicating that cooling remained detectable within the nonpainful range during the block. The change in CDT (baseline vs. block) was unrelated to the block duration (S2A Fig). Further, considering 23°C as the lower border of normal values for innocuous cold detection in the foot [39], we found no differences when comparing pain ratings (at $PAIN^{th}$) between participants with CDT above or below the lower border of innocuous CDT (S2B Fig).

Warm detection thresholds (WDTs) were not significantly different between baseline and block conditions, or between block and recovery conditions (Fig 2E). A significant correlation was found between the change in WDT and the duration of the nerve block (S2C Fig).

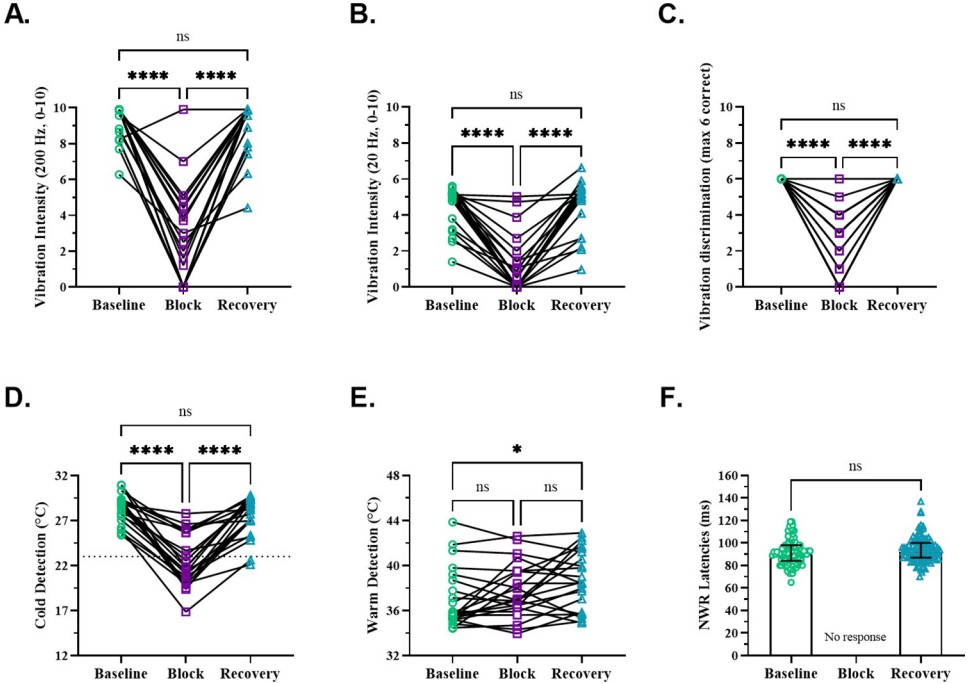

**Fig 2. Assessment of vibratory and thermal perception and NWR during nerve block. A-C**. Vibration intensity ratings for 200 Hz (A) and 20 Hz (B) declined significantly during the nerve block (*200 Hz*: baseline 9.9 (1.4), block 1.8 (4.4), recovery 9.9 (1.2), $f(2) = 32.38$, $p < 0.0001$; *20 Hz*: baseline 5.1 (1.6), block 0.3 (1.6), recovery 5.1 (1.7), $f(2) = 33.77$, $p < 0.0001$, post hoc test: ****$P < 0.0001$, ns > 0.9999, n = 22, Friedman test). Vibration discrimination (6 trials per condition) was also significantly impaired during the block (*3AFC*: baseline 6.0 (0.0), block 3.0 (3.0), recovery 6.0 (0.0), $f(2) = 40$, $p < 0.0001$, post hoc test: ****$P < 0.0001$, ns > 0.9999, n = 21, Friedman test). **D-E**. Cold detection thresholds (CDTs) significantly changed during the block (baseline 28.8 (1.9°C, block 21.6 (5.6°C, recovery 28.0 (2.8°C, $f(2) = 35.27$, $p < 0.0001$, post hoc test: ****$P < 0.0001$, ns = 0.395, n = 22, Friedman test). The dotted line at 23°C represents the lower border of normal values for innocuous cold detection, with cold pain emerging ≤10–14°C. Warm detection thresholds (WDTs) remained unchanged during the block but were elevated in the recovery condition compared to baseline (baseline 35.8 (3.6°C, block 37.0 (3.3°C, recovery 38.4 (5.5°C, $f(2) = 7.44$, $p = 0.024$, post hoc test: *$P = 0.031$, ns (baseline vs block) > 0.9999, ns (baseline vs recovery) = 0.150, n = 22, Friedman test). **F**. The reflex responses were completely abolished during the nerve block (NWR latencies: baseline 90.0 (14.0) ms, block 0.0 (0.0) ms, recovery 92.5 (13.0) ms, $p = 0.052$, U = 4113, n = 198, Mann Whitney test).

## NWR abolished by preferential Aβ-fiber block

During the nerve block, when the reflex was tested at $NWR^{th}$, all responses at their pre-block $NWR^{th}$ were abolished ([Fig 2F]). Despite further increases in current (up to 2 times the pre-block $NWR^{th}$ intensities) and/or prolonging of the pulse duration (extended to 1 ms), the reflex did not recover during the nerve block. This was true even for those participants (n = 5) whose block CDTs were within 1-3°C of their intact CDTs, yet no reflex responses were evoked.

## Reduced pain during preferential Aβ-fiber block

During the nerve block, pain ratings at the pre-block $PAIN^{th}$ dropped significantly, resulting in the complete abolition of pain in 14 out of 22 participants ([Fig 3A]). Reaction time measurements at the pre-block $PAIN^{th}$ were significantly delayed (baseline 258.8 ms, block 426.2 ms, n = 6), suggesting perception mediated via slower-conducting first-order afferents ([S3A Fig]). In four participants, pain ratings at $PAIN^{th}$ increased during the nerve block, an effect unrelated to the block duration ([S3B Fig]). Further, these four participants were not different from the others when comparing reflex latencies, pain ratings at $PAIN^{th}$ or $NWR^{th}$, vibration

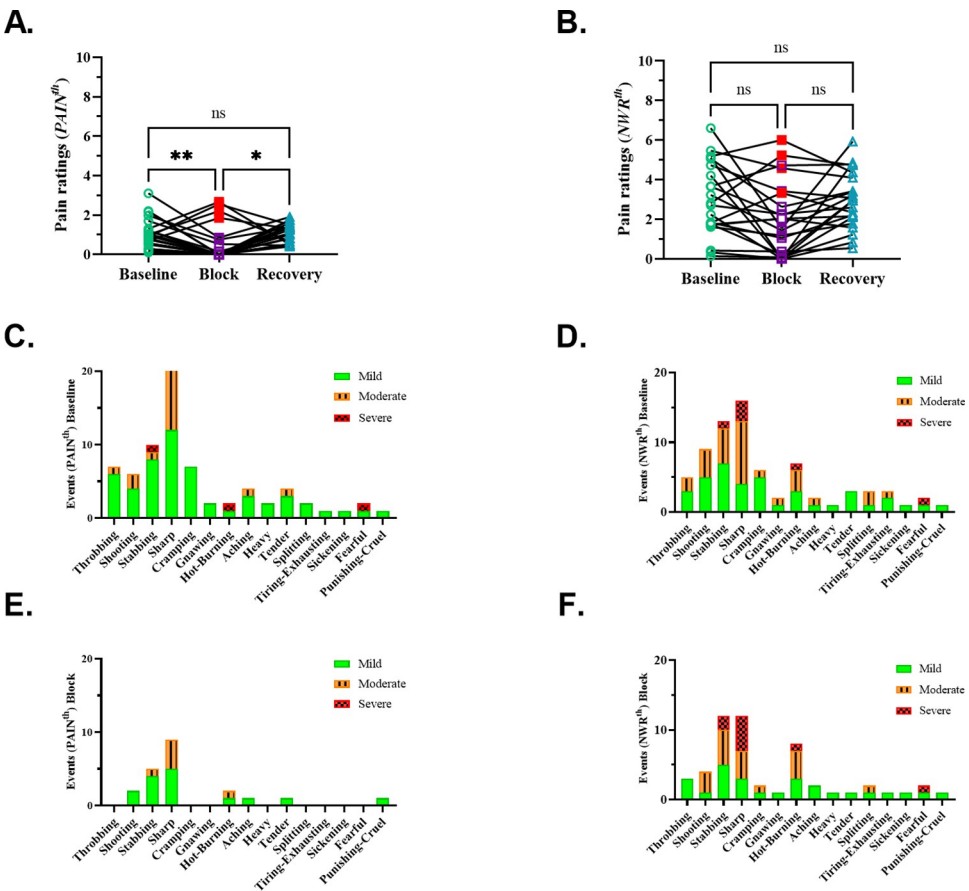

**Fig 3. Effect of nerve block on pain intensity and quality. A.** Reduction in pain ratings at $PAIN^{th}$ during nerve block. Pain ratings at the pre-block $PAIN^{th}$ were significantly reduced during the block (baseline 0.9 (0.7), block 0.0 (0.7), recovery 1.0 (0.9), f(2) = 11.55, p = 0.003, post hoc test: *P = 0.013, **P = 0.008, ns > 0.999, n = 22, Friedman test). Pain was completely abolished in 14 participants, greatly reduced in another 4, and increased in the remaining 4 (highlighted in red). **B.** Pain ratings at the pre-block $NWR^{th}$ did not significantly change across conditions (baseline 2.8 (3.1), block 1.8 (3.2), recovery 3.0 (2.2), f(2) = 6.181, p = 0.0331, post hoc test: ns (baseline vs block and baseline vs recovery) = 0.071, ns (baseline vs block) > 0.9999, n = 22, Friedman test). The 4 participants who showed an increase in $PAIN^{th}$ during nerve block in (a) are highlighted in red. **C-D**. Pain qualities at the pre-block $PAIN^{th}$ is shown on the left, while pain qualities at the pre-block $NWR^{th}$ is shown on the right. **E-F.** Pain qualities at block $PAIN^{th}$ is shown on the left and pain qualities at block $NWR^{th}$ is shown on the right. On each occasion, participants chose any number of descriptors and ranked their intensity as mild, moderate, or severe. Thus, the maximum number of "events" for each descriptor equals the number of participants (n = 22). The y-axis shows how many times a descriptor was chosen, and the x-axis shows the complete list of descriptors from the McGill short-form questionnaire.

sensibility, or temperature thresholds. They also did not differ based on NWR$^{th}$ or pulse duration (S3C–S3J Fig).

The most frequently chosen descriptors for characterizing pain quality at $PAIN^{th}$ were "sharp" and "stabbing" (Fig 3C). At $NWR^{th}$, "shooting" and "hot-burning" were also frequently chosen (Fig 3D). The proportion of descriptor intensity ranked as moderate and severe increased with increasing current from $PAIN^{th}$ to $NWR^{th}$ (mild-moderate-severe: 74-21-5% *to* 53-39-8%, respectively; Fig 3C and 3D).

During the block, while the NWR was abolished (Fig 2F), pain ratings at the pre-block $NWR^{th}$ did not differ from baseline levels. However, the overall occurrence of descriptors (and their corresponding intensity) reduced by 71% and 66% at $PAIN^{th}$ and $NWR^{th}$, respectively (Fig 3E and 3F).

In the control experiment, pain ratings at $PAIN^{th}$ and $NWR^{th}$, as well as NWR latencies, remained unchanged when the nerve block was applied to the contralateral leg (S4A–S4C Fig).

## Discussion

A preferential Aβ-fiber block significantly reduced pain and completely abolished NWR responses. The involvement of specific classes within the A-fiber population remains to be delineated. However, the abolition of $NWR^{th}$ responses during the block, despite using supra-threshold intensities in a condition in which cold perception, while reduced, was still detectable, invites speculation that Aβ nociceptors might be involved. Likewise, the reduction in $PAIN^{th}$ ratings during the block aligns with previous psychophysical findings indicating reduced mechanical pain perception in patients with selective Aβ deafferentation and normal mechanical pain perception in patients with selective small-fiber deafferentation [24, 40]. Moving forward, studying NWR in patients with selective large- or small-fiber deafferentation may provide valuable insights into distinct peripheral afferent contributions.

The recently discovered Aβ nociceptors in human skin are particularly well-suited to signal percepts and responses requiring rapid transmission of nociceptive information from the periphery [24–26]. In microneurography, intraneural stimulation of Aβ nociceptors produces painful percepts at the same current where intraneural stimulation of Aβ tactile afferents produces nonpainful percepts [24].

In the literature, the short-latency reflex component is considered non-nociceptive with the fast-conducting presumed "tactile" inputs thought to serve a role in posture correction or the inhibition of the late reflex response [10, 13, 20]. In the current study using intradermal electrical stimulation, we found no instances of dual NWR responses (e.g. two reflex responses within the same reflex recording time window). The NWR responses in our data had latencies ranging from 65 to 137 ms (mean latency: 91 ms), corresponding to a potential mix of RII and RIII latencies; however, they were rated as equally painful regardless of latency. The rapid return to baseline, following stimulation, allowed clear visual inspection of EMG recordings for short-latency responses. Since no NWR responses occurred prior to 50 ms, an analysis window 50–150 ms was adopted to include both short and long latency responses. Upon examining data from all individual reflex trials, only 9 out of 198 reflex responses (4.5%) were perceived as non-painful. These non-painful reflexes did not have the shortest latencies. The absence of pain in these instances might be due to a momentary shift in the participants' attention, as they never rated two consecutive reflex responses as non-painful.

In the current study, we used intradermal stimulation (needle electrodes), whereas the conventional approach is to use surface electrodes. In a previous study where surface electrical stimulation was used, 12.4% of the NWR responses had a dual component. Further, 14.2% of the NWR responses were rated as non-painful, and those had latencies ranging from 64 to 140 ms, with a median latency at 83.0 ms [22]. While one study found no difference between the two methods [41], another study, using both needle and surface electrodes, noted the absence of the early component of the reflex response (RII) during surface stimulation [11]. In that study, the sural nerve was stimulated perineural, unlike in our study where we were stimulating the reflex receptive field of the tibialis anterior muscle using needle electrodes inserted just under the skin of the foot sole. The targeted nerves (sural or tibial) or stimulation paradigms (duration and number of pulses) could be other factors determining if dual NWR responses are observed or not.

We used short-duration electrical pulses (0.2 ms) to preferentially activate larger-diameter fibers, based on the strength-duration relationship for electric excitation of myelinated axons [42–44], and to reduce the potential for irregular repetitive firing from primary afferent fibers

in response to prolonged stimulation [45]. In a few participants, the reflex was elicited using a longer pulse duration (1 ms), but this did not suggest activation of different nerve fibers, as latencies and pain ratings were not different from reflexes elicited at shorter pulse durations (S1A–S1D Fig). Consistent with earlier studies, and our pilot observations, using a single pulse failed to evoke NWR responses, indicating that temporal summation of repeated stimuli is required for reflex elicitation [7, 41, 46]. Typically, this need for multiple stimuli is overlooked when calculating conduction velocity in the afferent limb, as measurements are routinely made from the onset of the first pulse which usually fails to evoke an NWR. As NWR responses are typically elicited by trains of pulses separated by 2–5 ms, the conduction velocity of the afferent fibers contributing to the response could be underestimated, leading to bias towards the slower (Aδ) conduction range. Furthermore, the observations that NWR latencies are reduced by 3–4 ms during voluntary muscle contraction [17] and that motor neuronal response is enhanced for 50–150 ms following low-threshold electrical stimulation [47] indicate that the motor neuronal pool is a key determinant of the timing and amplitude of the NWR.

NWR relies on temporal summation driven by high-frequency repeated stimuli. In this context, the high impulse rates (up to 300 Hz) produced by Aβ nociceptors [24, 48] in response to noxious stimuli suggests that this class of afferent fiber is ideally suited for detection (and rapid relay) of information about noxious stimuli and to contribute to the generation of NWR. Indeed, the function of pain as a warning system necessitates the rapid transmission of information from the periphery to execute appropriate motor responses and meet behavioral requirements.

During the block, pain ratings at the pre-block $PAIN^{th}$ were significantly reduced, along with the frequency of pain descriptors and their corresponding intensities. The most frequently chosen pain descriptor "*sharp*" matches the percept evoked by selective activation of single Aβ nociceptors using low-current intraneural stimulation [24]. During the block, overall pain ratings at the pre-block $NWR^{th}$ did not change, but the frequency of pain descriptors and their corresponding intensities were greatly reduced. It may be that the increase in stimulus intensities from $PAIN^{th}$ to $NWR^{th}$ led to the activation of additional afferent types (Aδ and possibly C fibers), which might explain the persistence of pain at the pre-block $NWR^{th}$ during the block. This shift towards reliance on small-diameter inputs is reflected in the prolonged reaction times for electrical pain at $PAIN^{th}$ during the block (baseline 258.8 ms, block 426.2 ms), which now fall within the same range as reaction times for cold detection–a known Aδ function [49].

During the block, while pain ratings at pre-block $PAIN^{th}$ were reduced or abolished in most participants, in four of them, the pain was intensified. However, only the pain ratings, and not the NWR responses, differed from the other participants (S3C Fig).

During nerve block, CDT was significantly affected (reduced sensitivity), and detection thresholds are known to be variable. However, no difference was found in pain ratings at pre-block $PAIN^{th}$ between participants with reduced and normal cold sensitivity during the block. Importantly, during the block, despite a two-fold increase in pre-block $NWR^{th}$ and a five-fold increase in pulse duration, the reflex could not be evoked, even though all participants could still detect cooling in the nonpainful range (>10–14˚C) mediated by Aδ fibers. Warm sensibility, a function of C fibers, remained unaffected during the block (p > 0.999). WDT values were statistically elevated in the recovery condition compared to baseline, which could possibly be a consequence of increased blood flow to the limb upon cuff release, masking the participants' ability to detect warm temperatures. A longer waiting time for full recovery (>30 min) might have eliminated that difference.

Nociceptive responses can be modulated by another nociceptive stimulus in a phenomenon known as diffuse noxious inhibitory control (DNIC), where 'pain inhibits pain'. This is caused by inhibition in the spinal dorsal horn by nociceptive input from an adjacent part of the body [50]. The DNIC phenomenon has also been tested in relation to the NWR, resulting in increased NWR thresholds following exposure to noxious cold, heat, and muscular exercise [51]. To investigate potential DNIC effects on electrical pain and reflex, we performed control experiments with the blood pressure cuff applied to the contralateral leg. However, we found that pain and the NWR persisted ipsilaterally, indicating that the diminution of pain and the complete abolition of NWR were the result of a preferential loss of Aβ inputs when the cuff was applied to the ipsilateral leg rather than a DNIC effect. No participant reported pain from the nerve block, and although some found the pressure cuff uncomfortable, it was transient, and the experiment continued to completion.

NWR responses can also be modulated by cognitive factors, such as stress, anticipation, distraction, and emotions [4, 15, 52–54]. Although these aspects have been explored in other studies, they were beyond the scope of the current work. Nonetheless, certain measures were implemented; for instance, inter-stimulus intervals were varied between trials to prevent habituation, and care was taken to eliminate auditory and visual cues during testing. Our control experiment, involving a pressure cuff on the contralateral leg, further supports that the abolition of reflex and pain was primarily due to the blockade of peripheral input rather than top-down modulation. Our control experiment, with the pressure cuff on the contralateral leg, further supports that the abolition of reflex and pain were not primarily dependent on top-down modulation.

We made several important observations regarding pain and the NWR in our experimental conditions. The NWR consisted of a single response, evoked at stimulus intensities that were always painful, with no NWR thresholds observed below the pain threshold. During a preferential block of Aβ afferents, pain at the perception threshold was diminished, and the NWR was abolished. These results suggest the possible involvement of very fast-conducting afferents in pain perception and NWR signaling and may be relevant for understanding the functions of recently discovered human Aβ nociceptors.

## Supporting information

**S1 Fig.** A. Distribution of $NWR^{th}$ separated by pulse duration. B. NWR latencies separated by pulse duration. C-D. Pain ratings at $PAIN^{th}$ and $NWR^{th}$ separated by pulse duration across conditions.
(TIF)

**S2 Fig.** A. Comparison between the duration of nerve block and change in cooling sensitivity. B. Pain ratings at $PAIN^{th}$ between subjects with CDT below or above 23°C. C. Comparison between the duration of nerve block and change in warming sensitivity.
(TIF)

**S3 Fig.** A. Reaction time increased during nerve block. B. Duration of nerve block. C-E. Comparison of reflex latencies, $NWR^{th}$ and pulse duration between sensitized and normal subjects. F-H. Comparison of performance on vibratory tests during the block between sensitized and normal subjects. I-J. Comparison of performance on thermal tests during the block between sensitized and normal subjects.
(TIF)

**S4 Fig.** A. Pain ratings at $PAIN^{th}$ on the intact test site with a nerve block applied to the contralateral leg. B. Pain ratings at $NWR^{th}$ on the intact test site with a nerve block applied to the

contralateral leg. C. NWR latencies on the intact test site with a nerve block applied to the contralateral leg.
(TIF)

**S1 Database.**
(XLSX)

## Acknowledgments

We would like to thank Magnus Kronander for his helpful contributions to data compilation and analysis.

## Author Contributions

**Conceptualization:** Oumie Thorell, David Mahns, Jan Otto, Håkan Olausson, Saad Nagi.

**Data curation:** Oumie Thorell, Mats Svantesson.

**Formal analysis:** Oumie Thorell, David Mahns, Mats Svantesson, Saad Nagi.

**Investigation:** Oumie Thorell, Saad Nagi.

**Methodology:** Oumie Thorell, David Mahns, Jan Otto, Saad Nagi.

**Project administration:** Jaquette Liljencrantz, Håkan Olausson, Saad Nagi.

**Resources:** Håkan Olausson, Saad Nagi.

**Software:** Mats Svantesson.

**Supervision:** David Mahns, Jaquette Liljencrantz, Håkan Olausson, Saad Nagi.

**Validation:** David Mahns, Saad Nagi.

**Visualization:** Saad Nagi.

**Writing – original draft:** Oumie Thorell, Håkan Olausson, Saad Nagi.

**Writing – review & editing:** Oumie Thorell, David Mahns, Jan Otto, Jaquette Liljencrantz, Mats Svantesson, Håkan Olausson, Saad Nagi.

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
