## [Decision Letter · Decision Letter 0]

4 Jun 2024

PONE-D-24-18263Experimental nerve block study on painful withdrawal reflex responses in humansPLOS ONE

Dear Dr. Thorell,

Thank you for submitting your manuscript to PLOS ONE. After careful consideration, we feel that it has merit but does not fully meet PLOS ONE’s publication criteria as it currently stands. Therefore, we invite you to submit a revised version of the manuscript that addresses the points raised during the review process.

We look forward to receiving your revised manuscript.

Kind regards,

Tomoyoshi Komiyama, Ph.D

Academic Editor

PLOS ONE

Journal Requirements:

"This work was supported by the Swedish Research Council (S.S.N.),

https://www.vr.se/english.html

Knut and Alice Wallenberg Foundation (H.O.), https://kaw.wallenberg.org/en

ALF Grants, Region Östergötland (S.S.N.),https://www.researchweb.org/is/regionostergotland/anslag/anslagForskningsALF

Svenska Läkaresällskapet (S.S.N.), https://www.sls.se/

and Western Sydney University (D.A.M.), https://www.westernsydney.edu.au/schools/grs/scholarships/main_round_scholarships

The funders had no role in the study design, data collection and analysis, decision to publish, or preparation of the manuscript."

4. We notice that your supplementary figures are uploaded with the file type 'Figure'. Please amend the file type to 'Supporting Information'. Please ensure that each Supporting Information file has a legend listed in the manuscript after the references list.

Additional Editor Comments:

Dear Authors,

Your research proposed the possible involvement of very fast conducting afferents in pain perception and NWR signaling.

Additionally, you suggested that may be relevant for understanding the functions of recently discovered human Aβ nociceptors.

We thus have some questions and suggestions for the manuscript that you might consider.

I believe these comments will be very helpful in the revision of your study.

Tomoyoshi Komiyama

Reviewers' comments:

Reviewer's Responses to Questions

**Comments to the Author**

1. Is the manuscript technically sound, and do the data support the conclusions?

Reviewer #1: Partly

Reviewer #2: Yes

2. Has the statistical analysis been performed appropriately and rigorously? 

Reviewer #1: Yes

Reviewer #2: Yes

3. Have the authors made all data underlying the findings in their manuscript fully available?

Reviewer #1: Yes

Reviewer #2: Yes

4. Is the manuscript presented in an intelligible fashion and written in standard English?

Reviewer #1: Yes

Reviewer #2: Yes

5. Review Comments to the Author

Reviewer #1: Thorell et al. demonstrated that the effect of a preferential conduction block of Aβ fibers on pain perception and nociceptive withdrawal reflex signaling evoked by intradermal electrical stimulation in healthy participants.

Most of the study is well controlled, properly interpreted, and the data is clearly presented. A few suggestions for improvement are listed below.

1. Is it necessary to increase the number of participants more to obtain clearer results? Was there any confounding between age/gender and experimental results?

2. The aim of this (or future) work could be better defined, for instance to study interplay between Aβ and Aδ nerve activity under various (patho)physiological conditions.

3. The authors in the discussion section reported that “Intradermal electrodes, as used in the current study, are likely to stimulate the terminal branches of cutaneous afferents rather than the nerve endings themselves, resulting in a less synchronized afferent volley, perhaps resembling a more natural stimulus compared to surface electrodes”. I think it would be helpful to compare and contrast the differences due to this different approaches, as well as to discuss the feasibility of this latter approach.

Reviewer #2: The manuscript "Experimental nerve block study on painful withdrawal reflex responses in humans" presents the results of a study that sought to investigate the potential contributions of Abeta fibre input to pain and reflex signalling. Using an experimental nerve block to preferentially impede Abeta fibre contributions during nociceptive withdrawal reflex assessments as well as cold detection, the authors observed significant differences between the nerve block condition compared with the pre- and post-block conditions.

Overall, the manuscript is well written. The train of thought is easy to follow. The study is well motivated, and the results are presented in a clear and concise manner.

However, there are some concerns primarily relating to the methodology. Some choices the authors made warrant further justification or explanation. It should be quite easy for the authors to address those concerns and to make the appropriate changes, which should benefit their manuscript.

Please refer to the comments below for details.

Major comments:

Methods

line 27 (and 203)

It is mentioned that during the intact condition 198 NWR responses were recorded. Does this mean 9 per participant (as only 22 showed reflex responses)? How many elicitations were applied per participant? And how many were successful in obtaining a NWR response? I.e. what was the response rate? How many elicitations were applied in the other two conditions? This relevant information seems to be completely missing.

lines 104-108

Why was a pulse duration of 0.2ms chosen? It seems extremely short to elicit reflex responses and is not common in NWR studies. For instance, reference 16 uses the typical 1ms duration (see also reference 7). Using short pulse durations increases the risk of the stimulation not being perceived at low stimulation currents. The statement in the methods that, in cases were no reflexes could be elicited "a longer pulse duration of 1ms was used", acknowledges this issue (which affected 7 out of 22 participants, see lines 210-211). Therefore, please justify the chosen setting of 0.2ms or provide pertinent references.

Was it checked whether the choice of pulse duration and the obtained reflex threshold were significantly associated or not? Given the huge spread of obtained thresholds (visible in Fig. 1-B) one might want to make sure that this is not just due to the chosen pulse duration. Lines 211-213 only refer to potential differences in latencies but does not mention the threshold values, whose distribution is shown in Fig. S1-A.

line 128

Why was the reflex window set to 50-150ms post stimulus? According to reference 7, typical latencies for RII are between 40-60ms post stimulus. And according to 9, "a natural separation point between RII and RIII reflexes would be around 60ms".

Results

lines 273-275

Were these four participants different from the rest w.r.t. to NWR threshold stimulation current? It is mentioned in lines 381-382 that NWR responses did not differ between these four and the rest. But, no mention is made about the NWR threshold.

And in lines 205-207

it is mentioned that 90ms was seen as the cutoff for RII reflexes. So, why and how was the 50-150ms window assessed?

Discussion

lines 329-331 (and 109-111)

It has been observed that attention and distraction are associated to reflex responses (see Bjerre L. et al., "Dynamic tuning of human withdrawal reflex receptive fields during cognitive attention and distraction tasks", EJP 2011). Was attention or distraction of the participants, especially during reflex elicitations, controlled for in a systematic way? If so, please state how this was done. If not, please justify or elaborate.

Minor comments:

Abstract

line 29

misplaced "was"; it should read "...the NWR magnitude was..."

Methods:

line 115

" Current intensities were slowly increased in increments of 1-3 mA"

Was this systematically varied between participants? Why were different increments used? Please briefly explain this point.

lines 123-124

Participants had to fill in the short form of the McGill Pain Questionnaire a total of six times. Does this mean the procedure of threshold determination was performed six times in total (thrice for pain threshold and thrice for reflex threshold)?

line 129

Having a baseline window that extends up to the very moment of stimulus onset can prov difficult because of artefacts from the stimulus. Was this checked during data analysis?

again in line 165

What does "at leas 3 times" mean?

line 136 (and 173-174)

Was it checked whether participants felt pain from the ischemic nerve block, e.g. at the ankle, and how this might have influenced the results?

Fig. S3-A

Maybe indicate "Reaction Time (msec)" instead of just "Time (msec)" on the vertical axis.

General

The wording "current intensity" should not be used when referring to electricity because it simply is unscientific. An electrical current is measured in Ampère and refers to the local change in electrical charge per unit of time whereas an intensity refers to a power measurement (for instance in Watt) and -in electrodynamics- would be proportional to the square of the current. Thus, it is wrong to talk about "intensity" in the present context.

6. PLOS authors have the option to publish the peer review history of their article (what does this mean?). If published, this will include your full peer review and any attached files.

Reviewer #1: No

Reviewer #2: **Yes: **Alexandros Guekos

---

## [Author Response · Author response to Decision Letter 0]

18 Jul 2024

Reviewer 1#

1. Is it necessary to increase the number of participants more to obtain clearer results? Was there any confounding between age/gender and experimental results?

Detailed response:

In this study, our main finding was the complete abolition of the NWR response, observed in each participant. In the absence of pre-existing data for direct comparison, our initial power calculations were based on pain ratings, from a pilot study (n=7), at PAINth evoked by trans-cutaneous electrical stimulation (c.f., intradermal stimulation in this study), where we observed an effect size (f) of 0.255. We then used a 1-β error probability (power) of 0.80, and α error probability of 0.05, which gave a total sample size of 27. Post hoc power analysis, based on f(0.27), α(0.05), and a sample size of 25, gave a power of 82.6. Sample size and power calculations were performed in G* Power (open software, v3.1.9.7), which indicated enough power.

A correlation matrix showed no significant correlations between age, sex, NWRth and PAINth (with the lowest p value = 0.151 between age and NWRth). NWRth is not known to be affected by age or sex (Neziri, Andersen et al. 2010), and although PAINth has been reported to increase with age (Neri and Agazzani 1984), all participants in this study belonged to a relatively narrow age bracket (median age: 23.5 years, range: 18-39 years).

Corresponding changes to the manuscript:

• Lines 84-85: Twenty-five healthy participants (17 males and 8 females), aged between 18 and 39 years (median age: 23.5 years), took part in this study. 

• Lines 211-212: Detection, pain, or reflex thresholds did not show any correlation with the age or sex of participants.

2. The aim of this (or future) work could be better defined, for instance to study interplay between Aβ and Aδ nerve activity under various (patho)physiological conditions.

Detailed response:

While NWR has been extensively studied, including in various pathological conditions (Sandrini, Arrigo et al. 1993, Andersen, Finnerup et al. 2004, Banic, Petersen-Felix et al. 2004, Courtney, Lewek et al. 2009, Biurrun Manresa, Neziri et al. 2011, Ydrefors, Karlsson et al. 2020), the contribution of specific A-fiber classes remains poorly characterized. This is likely due to the difficulties inherent in isolating individual contributions within an intact system when using electrical stimulation alone, prompting us to use preferential nerve conduction blocks in the current study. Moving forward, studying patients with selective large or small fiber neuropathies may provide greater insights into the roles of specific A-fiber classes.

Corresponding changes to the manuscript:

• Lines 331-332: Moving forward, studying NWR in patients with selective large- or small-fiber deafferentation may provide valuable insights into distinct peripheral afferent contributions. 

3. The authors in the discussion section reported that “Intradermal electrodes, as used in the current study, are likely to stimulate the terminal branches of cutaneous afferents rather than the nerve endings themselves, resulting in a less synchronized afferent volley, perhaps resembling a more natural stimulus compared to surface electrodes”. I think it would be helpful to compare and contrast the differences due to these different approaches, as well as to discuss the feasibility of this latter approach.

Changes to the manuscript:

• Lines 357-361: While one study found no difference between the two methods (Meinck, Piesiur-Strehlow et al. 1981), another study using both needle and surface electrodes noted the absence of the early component of the reflex response (RII) during surface stimulation (de Willer 1977). However, when comparing the methods, in that study, the sural nerve was stimulated perineurally, whereas in the current study, we stimulated the reflex receptive field of the tibialis anterior muscle by inserting needle electrodes under the skin of the foot sole. 

The following text was removed from the manuscript to avoid confusion:

• Intradermal electrodes, as used in the current study, are likely to stimulate the terminal branches of cutaneous afferents rather than the nerve endings themselves, resulting in a less synchronized afferent volley, perhaps resembling a more natural stimulus compared to surface electrodes (Shahani and Young 1971). 

Reviewer #2

Major comments:

Methods

line 27 (and 203)

It is mentioned that during the intact condition 198 NWR responses were recorded. Does this mean 9 per participant (as only 22 showed reflex responses)? How many elicitations were applied per participant? And how many were successful in obtaining a NWR response? I.e. what was the response rate? How many elicitations were applied in the other two conditions? This relevant information seems to be completely missing.

Detailed response:

In our study, a distinction was made between “NWR responses”, which refers to all instances where an NWR was elicited, and “NWR thresholds”, which refers only to NWR responses at a specific current (mA). NWR thresholds (NWRth) were determined based on at least two successful trials at the same current. All participants had an NWR response, and each participant had a corresponding NWRth.

Corresponding changes to the manuscript:

• Lines 204-207: Under intact (baseline) conditions, 198 NWR responses were obtained from 22 out of 25 participants, ranging from 4 to 16 responses per participant. A reflex threshold (NWRth) was established for all 22 participants. Three participants were excluded because the full experiment could not be performed, resulting in insufficient data.

• Lines 216-221: During the block condition, the NWRth was tested at least 3 times using each participant's individual pre-block current. Further, suprathreshold intensities (up to twice the pre-block current) were tested during the block condition. This was to make sure that any effect on the NWR during the block was not merely due to a shift in NWRth, but rather indicated a completion abolition of the NWR. In the recovery condition, the NWR was considered recovered if it was elicited at least once at the pre-block current threshold. 

lines 104-108

Why was a pulse duration of 0.2ms chosen? It seems extremely short to elicit reflex responses and is not common in NWR studies. For instance, reference 16 uses the typical 1ms duration (see also reference 7). Using short pulse durations increases the risk of the stimulation not being perceived at low stimulation currents. The statement in the methods that, in cases where no reflexes could be elicited "a longer pulse duration of 1ms was used", acknowledges this issue (which affected 7 out of 22 participants, see lines 210-211). Therefore, please justify the chosen setting of 0.2ms or provide pertinent references.

Detailed response:

Although 1 ms duration is often used, shorter pulse durations have also been favored (Meinck, Kuster et al. 1985). We used closely spaced (5-mm) intradermal electrodes to apply short-duration electrical pulses (0.2 ms) to preferentially activate larger-diameter fibers, based on the strength-duration relationship for electric excitation of myelinated axons (Hill 1936, Szlavik and de Bruin 1999, Wesselink, Holsheimer et al. 1999). 

Participants clearly perceived the stimulus. The current intensity required to reach DETth was the lowest, followed by PAINth, and finally NWRth. All reflex thresholds occurred in response to a painful stimulus, with pain intensity ratings ranging from 0.3 to 6 on a VAS of 0-10.

Corresponding changes to the manuscript:

• Lines 102-105: In microneurography, short pulse durations have been shown to evoke a pain percept when Aβ nociceptors are selectively activated (Nagi, Marshall et al. 2019). Our approach, using needle electrodes, aimed to minimize current spread. Therefore, we chose the shortest pulse duration at which an NWR could still be elicited. 

• Lines 365-368: We used short-duration electrical pulses (0.2 ms) to preferentially activate larger-diameter fibers, based on the strength-duration relationship for electric excitation of myelinated axons (Hill 1936, Szlavik and de Bruin 1999, Wesselink, Holsheimer et al. 1999) and to reduce the potential for irregular repetitive firing from primary afferent fibers in response to prolonged stimulation (Meinck, Kuster et al. 1985).

Was it checked whether the choice of pulse duration and the obtained reflex threshold were significantly associated or not? Given the huge spread of obtained thresholds (visible in Fig. 1-B) one might want to make sure that this is not just due to the chosen pulse duration. Lines 211-213 only refer to potential differences in latencies but does not mention the threshold values, whose distribution is shown in Fig. S1-A.

Response:

No difference was found between NWRth at 0.2 and 1 ms. 

Corresponding changes to Supplementary 1:

• (NWRth 0.2: 12.0 (12.0), NWRth 1.0: 13 (8.0), p = 0.500, U = 42.5, n = 22, Mann-Whitney test). 

Corresponding changes to the manuscript:

• Lines 229-231: In a subset of participants (n = 7), the reflex could only be evoked using a 1-ms pulse duration. However, there were no statistical differences in NWRth, reflex latencies, or pain ratings based on pulse duration, hence the data were combined (Fig S1A-D). 

line 128

Why was the reflex window set to 50-150ms post stimulus? According to reference 7, typical latencies for RII are between 40-60ms post stimulus. And according to 9, "a natural separation point between RII and RIII reflexes would be around 60ms".

Detailed response:

The choice of reflex time analysis window depends on the focus of the study. Due to a clear baseline, and small stimulus artefacts, we were able to detect NWR responses within milliseconds of stimulus onset, although the shortest latency observed was 65 ms. The short-latency responses reported by Hugon (Hugon 1973) were following non-nociceptive stimulation. 

Corresponding changes to the manuscript:

• Lines 131-134: Following individual inspection of EMG recordings, a time analysis widow of 50-150ms was adopted. Z scores (> 1) were calculated as the difference between peak amplitude (50-150 ms post-stimulus onset) and the mean baseline amplitude (-0.15 to 0 ms relative to the stimulus onset), divided by the standard deviation of baseline EMG activity.

• Lines 345-351: The rapid return to baseline, following stimulation, allowed clear visual inspection of EMG recordings for short-latency responses. Since no NWR responses occurred prior to 50 ms, an analysis window of 50-150 ms was adopted to include both short and long latency responses. Upon examining data from all individual reflex trials, only 9 out of 198 reflex responses (4.5%) were perceived as non-painful. These non-painful reflexes did not have the shortest latencies. The absence of pain in these instances might be due to a momentary shift in the participants’ attention, as they never rated two consecutive reflex responses as non-painful. 

Results

lines 273-275

Were these four participants different from the rest w.r.t. to NWR threshold stimulation current? It is mentioned in lines 381-382 that NWR responses did not differ between these four and the rest. But no mention is made about the NWR threshold.

Changes to the manuscript:

• Lines 290-293: Further, these four participants were not different from the others when comparing reflex latencies, pain ratings at PAINth or NWRth, vibration sensibility, or temperature thresholds. They also did not differ based on NWRth or pulse duration (Fig S3C-J).

Following figures have been added to Supplementary 3 (D and E): 

Corresponding changes to Supplementary 3:

• C-E. Comparison of reflex latencies, NWRth and pulse duration between sensitized and normal subjects. To simplify, subjects with an increased PAINth during the block are referred to as ‘sensitized’ while subjects who did not have an increase are referred to as ‘normal’. NWR latencies did not differ between sensitized (n=4) and normal (n=18) subjects (Sensitized: baseline 89.6 (9.7) ms, block 0.0 (0.0) ms, recovery: 93.6 (4.8) ms; Normal: baseline 91.8 (11.0) ms, block 0.0 (0.0) ms, recovery 93.9 (12.7) ms, f(2) = 42.79, p < 0.001, post hoc test: p(baseline & recovery) > 0.999, n = 22, Kruskal-Wallis test). NWRth and pulse durations were not different in the sensitized subjects compared to normal subjects (NWRth Sensitized: 12 (10.0), Normal: 20 (7.5), p = 0.137, U = 18, n = 22, Mann-Whitney test) (Pulse duration Sensitized: 0.6 (0.8), Normal: 0.2 (0.8), p = 0.565, U = 28, n = 22, Mann-Whitney test).

And in lines 205-207

it is mentioned that 90ms was seen as the cutoff for RII reflexes. So, why and how was the 50-150ms window assessed?

Changes to the manuscript:

• Lines 224-225: Out of these, 80 (40.4%) NWR responses had latencies under 90 ms, a cutoff defining onset of RIII reflexes, as used in prior studies (e.g. Sandrini, Arrigo et al. 1993, Ruscheweyh, Kreusch et al. 2011, Ydrefors, Karlsson et al. 2020).

Discussion

lines 329-331 (and 109-111)

It has been observed that attention and distraction are associated to reflex responses (see Bjerre L. et al., "Dynamic tuning of human withdrawal reflex receptive fields during cognitive attention and distraction tasks", EJP 2011). Was attention or distraction of the participants, especially during reflex elicitations, controlled for in a systematic way? If so, please state how this was done. If not, please justify or elaborate.

Detailed response:

Cognitive influences on reflexes have been investigated in other studies, including the one cited by the reviewer. Our focus here, however, was on the peripheral afferent contributions; nonetheless, to the extent possible within the scope of this work, some considerations were made. For instance, interstimulus intervals were varied from trial to trial (at least >6 s) to prevent habituation and/or cognitive suppression of the reflex response. Also, to avoid visual and auditory cues, a partition was placed between the participant and the experimenter, and a silent mouse was used to trigger the stimuli. It has indeed been shown that habituation can inhibit reflex responses (Dimitrijevic, Faganel et al. 1972), while anticipation can facilitate them (Willer 1980).

Corresponding changes to the manuscript:

• Lines 428-436: NWR responses can also be modulated by cognitive factors, such as stress, anticipation, distraction, and emotions (Willer, Boureau et al. 1979, Willer 1980, Sandrini, Milanov et al. 2000, Ruscheweyh, Kreusch et al. 2011, Toledo, Vore et al. 2024). Although these aspects have been explored in other studies, they were beyond the scope of the current work. Nonetheless, certain measures were implemented; for instance, inter-stimulus intervals were varied between trials to prevent habituation, and care was taken to eliminate auditory and visual cues during testing. Our control experiment, involving a pressure cuff on the contralateral leg, further supports that the abolition of reflex and pain was primarily due to the blockade of peripheral input rather than top-down modulation. 

We have also changed the order of paragraphs in the Discussion to improve flow. The paragraph discussing cognitive factors now directly follows the one on spinal gating (diffuse noxious inhibitory control).

Minor comments:

Abstract

line 29

misplaced "was"; it should read "...the NWR magnitude was..."

Changes to the manuscript:

• Lines 29-30: The current required to elicit the NWR was higher than the perceptual pain threshold, indicating that NWR did not occur before pain was felt. 

Methods:

line 115

"Current intensities were slowly increased in increments of 1-3 mA"

Was this systematically varied between participants? Why were different increments used? Please briefly explain this point.

Detailed response:

The use of different increments is a common practice in threshold testing, even more so when stimulations are manually controlled, like in the current study. 

Corresponding changes to the manuscript:

• Lines 115-116: Current was slowly increased and decreased in increments of 1-3 mA until the first (nonpainful) sensation was reported.

lines 123-124

Participants had to fill in the short form of the McGill Pa

---

## [Decision Letter · Decision Letter 1]

6 Aug 2024

Experimental nerve block study on painful withdrawal reflex responses in humans

PONE-D-24-18263R1

Dear Dr. Thorell,

We’re pleased to inform you that your manuscript has been judged scientifically suitable for publication and will be formally accepted for publication once it meets all outstanding technical requirements.

Kind regards,

Tomoyoshi Komiyama, Ph.D

Academic Editor

PLOS ONE

Additional Editor Comments (optional):

Dear authors,

Thank you for submitting your revised manuscript.

It was much easier to understand than the original manuscript.

I am satisfied with the responses and the edits, so I am happy to accept your study.

You have satisfactorily addressed the comments from the two reviewers.

Therefore, I have no further suggestions.

I believe this manuscript will satiate the reader's interest.

Tomoyoshi Komiyama

Reviewers' comments:

Reviewer's Responses to Questions

**Comments to the Author**

1. If the authors have adequately addressed your comments raised in a previous round of review and you feel that this manuscript is now acceptable for publication, you may indicate that here to bypass the “Comments to the Author” section, enter your conflict of interest statement in the “Confidential to Editor” section, and submit your "Accept" recommendation.

Reviewer #1: All comments have been addressed

2. Is the manuscript technically sound, and do the data support the conclusions?

Reviewer #1: Yes

3. Has the statistical analysis been performed appropriately and rigorously? 

Reviewer #1: Yes

4. Have the authors made all data underlying the findings in their manuscript fully available?

Reviewer #1: Yes

5. Is the manuscript presented in an intelligible fashion and written in standard English?

Reviewer #1: Yes

6. Review Comments to the Author

Reviewer #1: The author provided appropriate explanations and corrections to all questions. Therefore, this paper is considered for publication in PLOSOne.

7. PLOS authors have the option to publish the peer review history of their article (what does this mean?). If published, this will include your full peer review and any attached files.

Reviewer #1: No

---

## [Editor Report · Acceptance letter]

8 Aug 2024

PONE-D-24-18263R1 

PLOS ONE

Dear Dr. Thorell, 

I'm pleased to inform you that your manuscript has been deemed suitable for publication in PLOS ONE. Congratulations! Your manuscript is now being handed over to our production team.

Kind regards, 

on behalf of

Dr. Tomoyoshi Komiyama 

Academic Editor

PLOS ONE